# OmniDrive: Towards Unified Next-Gen Controllable Multi-View Driving Video Generation with LLM-Guided World Model

## Abstract

Recent diffusion-based world models can synthesize multi-camera driving videos, yet they still suffer from geometric drift between views, degrading perception, prediction and planning. We introduce OminiDrive, the first unified model that jointly compresses, generates and modulates all camera streams to deliver realistic, controllable and view-consistent driving videos. A DiT backbone operates in a shared latent manifold obtained by multi-view variational compression; within this space a consistency-aware denoiser injects correlated noise and aligns view-dependent coordinates at every diffusion step. Heterogeneous control signals—vehicle trajectory, ego pose and scene semantics—are fused through lightweight latent modulation layers, thus steering generation without extra inference cost. By reasoning over a single, view-homogeneous token grid, OminiDrive preserves both spatial coherence and temporal fidelity. Experiments on nuScenes and Waymo datasets show state-of-the-art view consistency and video quality, and the synthesized data significantly improves the performance of downstream perception models. The project is available at https://iclr2026sub.github.io/OminiDrive/.

## 1 Introduction

Generative world models Gao et al. (2023; 2024a); Wang et al. (2024a); Zhao et al. (2025b); Wen et al. (2024); Kim et al. (2021); Mei et al. (2024); Zhao et al. (2025a) have rapidly become a linchpin of autonomous-driving research. By amalgamating 3D VAEs Yang et al. (2024b); Kong et al. (2024) with DiT backbones Esser et al. (2024a) and accelerated by flow-matching samplers Lipman et al. (2022), modern diffusion systems now deliver minute-long, photorealistic, *controllable* simulations at automotive scale. Such synthetic corpora markedly curtail data-collection costs while enabling exhaustive, closed-loop evaluation of perception and planning stacks Yang et al. (2024a).

Despite this progress, two fundamental obstacles remain. **(i) Multi-view inconsistency.** Prevailing pipelines compress each of the six camera streams independently, Cross-view communication is therefore postponed to diffusion time via ad-hoc cross-attention Gao et al. (2024a); Wang et al. (2024a), leaving the latent space fragmented and geometrically discordant. **(ii) Heterogeneous control injection.** Driving simulators must reconcile spatially aligned geometric cues (HD-maps, trajectories, camera extrinsics) with global semantic cues (text or style prompts). Existing architectures attach disjoint modules—ControlNet-like branches for geometry, cross-attention adapters for semantics—so temporal synchrony and spatial anchoring are frequently lost, degrading fine-grained controllability.

To address the two challenges outlined above, we introduce *OminiDrive*, the first unified framework specifically designed for multi-camera driving video generation that resolves both issues through **Unified Compression** and **Unified Controllable Generation**. Unified Compression performs an early-fusion encoding of the six camera streams, allowing inter-view geometry to be inferred before any latent is produced and thereby eradicating cross-camera inconsistencies. Unified Controllable Generation then harnesses an LLM-assisted MM-DiT Esser et al. (2024b) that ingests a joint sequence of video latents, linguistic prompts distilled by the LLM, and geometric cues, so that spatially local conditions and global semantics are negotiated within one coherent representation. Working

Table 1: OminiDrive offers the most comprehensive control among the leading driving video generation models.

| Model | Supported Control Conditions | | | | | |
|---|---|---|---|---|---|---|
| | Traj. | 3D Box | HD Map | Text | Cam. | Img Ref. |
| DriveGANKim et al. (2021) | × | × | × | × | ✓ | × |
| Drive-WMWang et al. (2024b) | ✓ | × | × | × | × | × |
| Gen-ADZheng et al. (2024) | ✓ | × | × | ✓ | × | × |
| VistaGao et al. (2024b) | ✓ | × | × | ✓ | ✓ | × |
| GAIA2Russell et al. (2025) | × | ✓ | × | ✓ | ✓ | ✓ |
| **OminiDrive (ours)** | ✓ | ✓ | ✓ | ✓ | ✓ | ✓ |

in concert, these two designs furnish OminiDrive with both rigorous cross-view consistency and fine-grained, flexible control.

We fine-tune the public HUNYUAN-3D VAE Kong et al. (2024) on $1,500$ h of NUSCENES+ WAYMO footage and train the diffusion backbone via a three-stage curriculum. Empirically, *OminiDrive* attains sharper imagery, stronger geometric alignment, and closer adherence to control signals than competing systems. We contend that this unified architecture foreshadows a new paradigm for scalable, high-fidelity simulation of autonomous-driving scenarios. Overall, the contributions of this work are three-fold:

- We propose **OminiDrive**, whose Unified Compression eliminates inter-camera drift and whose LLM-guided Unified Controllable Generation achieves coherent fine-grained control.
- We devise an efficient training recipe for OminiDrive that combines lightweight VAE fine-tuning with a progressive diffusion curriculum, enabling minute-long, high-resolution synthesis.
- Extensive experiments demonstrate OminiDrive's clear advantages in visual quality, cross-view coherence, and controllability, establishing a strong foundation for future research in unified world modelling for autonomous driving.

## 2 RELATED WORK

### 2.1 GENERATIVE AND CONTROLLABLE WORLD MODELS

Embodied-AI simulators now use generative world models that roll out long-horizon video from ego actions. Diffusion rules the field: (i) UNet pipelines denoise 2D frames with 3D kernels or attention (MyGo, MagicDrive, DriveScapeYao et al. (2024); Gao et al. (2023); Wu et al. (2024)), lightweight yet prone to spatio-temporal drift; (ii) DiT transformers capture global structure but require heavy memory, forcing staged training or token pruning (DiVE, DiffusionDriveJiang et al. (2024); Liao et al. (2025)). Autoregressive renderers such as DriveGAN and DriveDreamerKim et al. (2021); Wang et al. (2024a) plan trajectories explicitly but accumulate Markov error, limiting high-res multi-view roll-outs. Control has progressed from single ControlNet branches to shared latent spaces that fuse geometry, identity, time, audio, and text—CineMaster and HunyuanVideoWang et al. (2025); Kong et al. (2024) typify the "train once, reuse everywhere" ethos. Driving models must also obey semantic cues (weather, style) and pixel-level geometry from HD maps, trajectories, and multi-camera rigs; early controllers like MagicMotion and MotionCtrlLi et al. (2025); Wang et al. (2024c) merely bolt such hints onto generic pipelines, support at most two views, and lack six-camera evaluation, leaving robust multi-view controllability unresolved.

### 2.2 MULTI-VIEW CONSISTENCY CONTROL

While single-camera realism has improved rapidly, synthesising temporally long and view-consistent surround video remains challenging. Existing strategies can be categorised into three

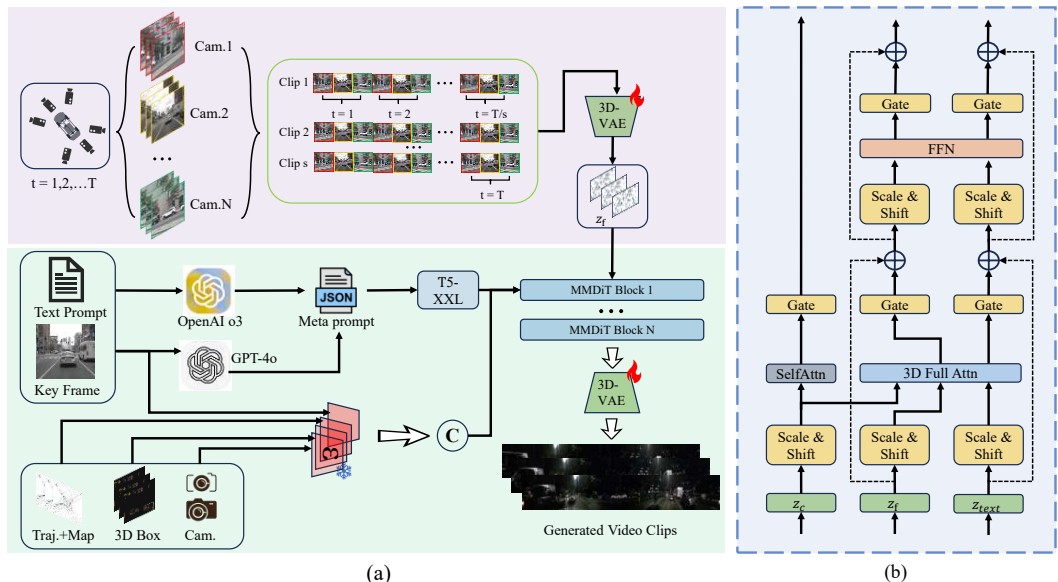

(a)  (b)

Figure 1: **Overview architecture of OminiDrive.** (a) The top part (purple background) is the Unified Compression module, which improves previous per-view compression methods into joint multi-view compression. The bottom part (light green background) is the Unified Controllable Generation module. Based on a unified conditional encoding scheme, OminiDrive provides a unified control mechanism for both spatially-aligned and non-spatially-aligned signals. (b) shows the improved MMDiT block used in OminiDrive, where 3D Full Attention jointly processes the latent representations of frames, text, and control signals.

research trajectories. Latent-space sharing encodes every view independently and aggregates the latents into a scene vector, as in the GAIA seriesRussell et al. (2025) and DreamFusionPoole et al. (2022); although lightweight, this late fusion cannot fully suppress inter-camera drift. Geometry-aware modelling injects explicit projective cues—camera matrices, epipolar correspondences or 3D neural fields—into the training objective, a paradigm embraced by MyGoYao et al. (2024), Drive-DreamerZhao et al. (2025b) and NVS-DiffusionYou et al. (2024). These methods deliver strong consistency but demand precise extrinsics, heavy supervision and long training schedules. A third line, view-aware attention, keeps the pipeline extrinsic-free by weaving cross-camera attention directly into the Transformer, as pursued by VideoComposerWang et al. (2023), DiVEJiang et al. (2024) and UniMLVGChen et al. (2024); yet the mechanism still operates on per-view latents and incurs a cubic cost in sequence length. More recent efforts such as MVDiffusionDeng et al. (2023) and Vivid-ZooLi et al. (2024a) reinforce these schemes with epipolar masks or 2D/3D alignment, but quantitative evidence indicates that repairing inconsistencies only at decoding is insufficient—texture mismatch and illumination drift persist.

# 3 METHODOLOGY

## 3.1 PRELIMINARIES

A 3-D variational auto-encoder (VAE) furnishes a smooth, hence differentiable, latent manifold, whereas Diffusion Transformers (DiT) equipped with *conditional flow matching* (CFM) enable few-step, precisely controlled synthesis. We summarise both components before introducing our unified formulation.

### 3.1.1 LATENT COMPRESSION WITH A 3D VAE

For a six-camera driving video we denote the raw tensor by $\mathbf{x} = \{x_{b,n,t} \in \mathbb{R}^{C \times H \times W}\}_{b=1,n=1,t=1}^{B,N,T}$, where $b$ indexes the batch, $n$ the camera, and $t$ the frame. The encoder produces a latent field

$$\mathbf{z}_0 = E_\phi(\mathbf{x}) + \boldsymbol{\sigma} \odot \boldsymbol{\epsilon}, \quad \boldsymbol{\epsilon} \sim \mathcal{N}(\mathbf{0}, \mathbf{I}), \qquad (1)$$

Table 2: **Quantitative comparison on the NUSCENES validation set.** ↑/↓ denote "higher is better" / "lower is better". A dash indicates that the model does not support the corresponding control. The best score is typeset in **bold**; the second best, when available, is underlined. Results marked with "*" are copied from the respective original papers because we could not reproduce them, and the authors did not disclose the exact resolution and frame length used.

| Model | Multi-view | Image Quality | | | Video Quality | | | |
|---|---|---|---|---|---|---|---|---|
| | | FID↓ | PSNR↑ | IQ↑ | FVD↓ | TF↑ | AQ↑ | Diversity↑ |
| MagicDriveDiT$_{\text{arXiv25}}$Gao et al. (2024a) | ✓ | 10.89 | 30.99 | 51.7% | 64.81 | 92.1% | 50.6% | 29.5% |
| DriveDreamer-2$_{\text{AAAI25}}$Zhao et al. (2025b) | ✓ | 14.32 | 29.89 | 50.6% | 55.70 | 95.2% | 51.4% | 33.1% |
| UniMLVG$_{\text{arXiv25}}$Chen et al. (2024) | ✓ | **5.80** | 31.04 | 57.7% | **36.10** | 95.0% | **55.6%** | 27.4% |
| Drive-WM$_{\text{CVPR24}}$Wang et al. (2024b) | ✓ | 25.88 | 26.91 | 49.2% | 122.70 | 86.3% | 44.1% | 37.9% |
| Drivescape*$_{\text{arXiv24}}$Wu et al. (2024) | ✓ | 8.34 | – | – | 76.39 | – | – | – |
| DiVE*$_{\text{arXiv24}}$Jiang et al. (2024) | ✓ | – | – | 51.82% | 94.60 | – | – | – |
| Delphi*$_{\text{arXiv24}}$Ma et al. (2024) | ✓ | 15.08 | – | – | 113.50 | – | – | – |
| Vista$_{\text{NIPS24}}$Gao et al. (2024b) | ✕ | 8.82 | 29.19 | 49.1% | 92.32 | 90.5% | 52.1% | 34.5% |
| Panacea$_{\text{CVPR24}}$Wen et al. (2024) | ✓ | 14.91 | 30.01 | 50.8% | 244.00 | 93.2% | 41.5% | 34.1% |
| DriveGAN$_{\text{CVPR21}}$Kim et al. (2021) | ✕ | 31.79 | 24.32 | 37.1% | 502.30 | 94.4% | 43.2% | **38.8%** |
| **OminiDrive (ours)** | ✓ | 8.01 | **31.15** | **59.5%** | 45.75 | **97.0%** | 53.4% | 33.7% |

which the decoder reconstructs as $\hat{\mathbf{x}} = D_{\boldsymbol{\theta}}(\mathbf{z}_0)$. Training maximises the evidence lower bound

$$\mathcal{L}_{\text{ELBO}} = \mathbb{E}_{q_{\boldsymbol{\phi}}}\big[\log p_{\boldsymbol{\psi}}(\mathbf{x}\,|\,\mathbf{z}_0)\big] - D_{\text{KL}}\big(q_{\boldsymbol{\phi}}(\mathbf{z}_0\,|\,\mathbf{x})\|\mathcal{N}(\mathbf{0},\mathbf{I})\big). \tag{2}$$

The convolutional encoder collapses redundant spatio-temporal structure, bending the high-curvature data manifold $\mathcal{M}_{\mathbf{x}}$ into a near-Euclidean latent manifold $\mathcal{M}_{\mathbf{z}}$ whose residual noise is well approximated by a Gaussian—an ideal substrate for deterministic flow integration.

### 3.1.2 CONTROLLABLE LATENT DIFFUSION VIA CONDITIONAL FLOW MATCHING

Let $\mathbf{z}_1 \sim \mathcal{N}(\mathbf{0},\mathbf{I})$ be standard latent noise and let

$$\mathbf{z}_t = (1-t)\mathbf{z}_0 + t\mathbf{z}_1, \quad t \sim \mathcal{U}[0,1], \tag{3}$$

denote the linear data-noise path proposed by rectified flow Lipman et al. (2022). The oracle velocity along this path is $v^{\star}(\mathbf{z}_t, t) = \mathbf{z}_1 - \mathbf{z}_0$, independent of $t$. A learnable predictor $v_{\boldsymbol{\theta}}$ is trained with the conditional flow-matching loss

$$\mathcal{L}_{\text{CFM}} = \mathbb{E}_{\mathbf{z}_0,\mathbf{z}_1,t,\mathbf{c}}\big\|v_{\boldsymbol{\theta}}(\mathbf{z}_t,t,\mathbf{c}) - (\mathbf{z}_1 - \mathbf{z}_0)\big\|_2^2, \tag{4}$$

where $\mathbf{c}$ denotes external controls (text, HD-map, *etc.*). At convergence, $v_{\boldsymbol{\theta}} \approx v^{\star}$ and the latent satisfies the probability-flow ODE

$$\frac{d\mathbf{z}_t}{dt} = v^{\star}(\mathbf{z}_t, t). \tag{5}$$

A DiTEsser et al. (2024a) backbone conditions on $v_{\boldsymbol{\theta}}$ using shared or offset positional indices that softly couple the six camera views. Deterministic integration from $t=1$ to $t=0$ yields the controlled sample $\mathbf{z}_0 = \mathbf{z}_1 + \int_1^0 v_{\boldsymbol{\theta}}(\mathbf{z}_t, t, \mathbf{c})\, dt$.

## 3.2 UNIFIED COMPRESSION

**View–time permutation.** To impose geometric coherence *before* encoding, we collapse view and time into one pseudo-temporal axis. Formally, for every pair $(n,t)$ we define the permutation

$$\Pi : (n,t) \longmapsto \tilde{t} = (n-1)T + t, \qquad \tilde{T} = NT, \tag{6}$$

and reorder the video as $\tilde{\mathbf{x}} = \{x_{b,\tilde{t}} \equiv x_{b,n,t}\}_{\tilde{t}=1}^{\tilde{T}}$. The permuted sequence is fed to the 3D encoder of Sec. 3.1, yielding latents $\mathbf{z}_0 \in \mathbb{R}^{B \times \tilde{T}' \times H' \times W'}$. Because $\Pi$ is lossless, the same ELBO in equation 2 applies.

**Why it works.** Adjacent indices along $\tilde{t}$ correspond to different cameras at the *same* physical instant; a single 3D convolution therefore "sees" all views concurrently, converting cross-view geometry into local temporal context. The inter-camera variance $\sigma_{\text{inter}}^2$ is effectively averaged over the kernel width $r_t$, reducing inconsistency without modifying the encoder's Lipschitz bound or requiring new parameters—pretrained VAE weights transfer verbatim to any camera count.

**Interface to generation.** Since the permutation leaves the latent metric unchanged, the flow ODE in equation 5 still enjoys few-step integration. The resulting tensor is concatenated with textual and geometric controls to form the single, aligned token sequence consumed by our *Unified Controllable Generation* module (Sec. 3.3).

### 3.3 UNIFIED CONTROLLABLE GENERATION

After the view–time permutation of Sec. 3.2, the six–camera footage has been compressed into a single pseudo-temporal latent tensor $\mathbf{z}_0 \in \mathbb{R}^{B \times \tilde{T}' \times H' \times W'}$. We now describe how *OminiDrive* realises fine-grained yet *unified* control on top of a Multi-Modal DiT (MM-DiT) backbone Esser et al. (2024b).

#### 3.3.1 LATENT TOKENS AND NOISE INJECTION

The clean latent $\mathbf{z}_0$ is perturbed along the linear path of equation 3 to obtain $\mathbf{z}_t$. A $k_t \times k_h \times k_w$ 3D convolutional *patchify* operator converts $\mathbf{z}_t$ into a set of $L$ tokens,

$$\mathbf{X} = \left\{ x_\ell \in \mathbb{R}^d \mid \ell = 1, \ldots, L \right\}, \qquad L = \frac{\tilde{T}'}{k_t} \cdot \frac{H'}{k_h} \cdot \frac{W'}{k_w}. \tag{7}$$

Each token is endowed with a 3D rotary positional embedding $\boldsymbol{\pi}(t, i, j)$ derived from its grid coordinate $(t, i, j)$.

#### 3.3.2 UNIFIED ENCODING OF CONTROL CONDITIONS

**(a) Semantic control $\mathbf{C}^{\text{sem}}$.** Non-aligned semantic guidance comprises text, style, and camera pose. The user prompt $p_{\text{usr}}$ is first rewritten by an LLM into a concise, disambiguated sentence $p_{\text{dense}}$. The dense sentence is serialised into a lightweight JSON object that explicitly labels *actors*, *actions*, and *attributes*. This structured representation (i) yields deterministic token boundaries, (ii) removes linguistic redundancy before CLIP encoding, and (iii) allows downstream modules to address individual tags, improving controllability in ablation (Sec. 4.4).

Encoding $p_{\text{dense}}$ with CLIP-ViT/L Radford et al. (2021) gives a feature matrix $\mathbf{e}_{\text{text}} \in \mathbb{R}^{M \times 768}$ which is linearly projected to $\mathbf{C}^{\text{text}} \in \mathbb{R}^{M \times d}$. Visual style is supplied by a key frame $\mathbf{I}_0$; passing $\mathbf{I}_0$ through the shared VAE and global-average pooling yields a style token $\mathbf{C}^{\text{sty}} \in \mathbb{R}^{1 \times d}$. Camera extrinsics $(R_c, t_c)$ are mapped to a *6-D Plücker ray* $\mathbf{r}_c \in \mathbb{R}^6$ and then to a pose token $\mathbf{C}^{\text{cam}} \in \mathbb{R}^{1 \times d}$ via a two-layer MLP. Concatenation forms $\mathbf{C}^{\text{sem}} = [\mathbf{C}^{\text{text}}; \mathbf{C}^{\text{sty}}; \mathbf{C}^{\text{cam}}] \in \mathbb{R}^{M_{\text{sem}} \times d}$. All semantic tokens are shifted by a constant offset $(\Delta_i, 0)$ along the spatial grid ($\Delta_i = 32$) so that they never collide with visual-grid indices.

**(b) Geometric control $\mathbf{C}^{\text{geo}}$.** Aligned geometric cues come from the HD-map $\mathbf{M}_t$, the set of 3D boxes $\mathbf{B}_t$, and the ego trajectory $\mathbf{Tr}_t$. They are rasterised into a sparse RGB image $\mathbf{I}_t^{\text{geo}}$, encoded by the same VAE, and patchified to tokens $\mathbf{C}^{\text{geo}} = \left\{ c_{t,i,j}^{\text{geo}} \in \mathbb{R}^d \right\}$, which *share* the spatial indices $(t, i, j)$ with the latent tokens in equation 7, guaranteeing pixel-level alignment.

**(c) Temporal tokens $\mathbf{C}^{\text{tmp}}$.** To model long-range dynamics we embed the normalised timestamp $\tau_t = t/\tilde{T}'$ with a 1-D sinusoid $\psi(\tau_t) \in \mathbb{R}^{d_\tau}$ and project it to $d$ dimensions, yielding $\mathbf{C}^{\text{tmp}} \in \mathbb{R}^{\tilde{T}' \times d}$. The function symbol $\psi$ avoids confusion with the bias parameter introduced below.

#### 3.3.3 UNIFIED SEQUENCE AND INTERACTION WITH MM-DIT

The four token sets are concatenated

$$\mathbf{S} = \left[ \mathbf{X}; \mathbf{C}^{\text{sem}}; \mathbf{C}^{\text{geo}}; \mathbf{C}^{\text{tmp}} \right] \in \mathbb{R}^{(L + M_{\text{c}}) \times d}, \tag{8}$$

Table 3: **Multi-view consistency and controllability results on the NUSCENES validation set.** ↑/↓ denote "higher is better" / "lower is better". A dash indicates that the model does not support the corresponding control.

| Model | Multi-view Consistency | | | | | Controllability | | | |
|---|---|---|---|---|---|---|---|---|---|
| | Foreground Consistency | | Background Consistency | | | Geometric Control | | Semantic Control | |
| | SC↑ | MS↑ | PC↑ | BC↑ | OC↑ | mAP↑ | mIoU↑ | Scene↑ | AS↑ |
| MagicDriveDiT$_{arXiv25}$Gao et al. (2024a) | 91.3% | 82.9% | 62.8% | 92.6% | 18.5% | 18.17 | **20.40** | 49.1% | 8.6% |
| DriveDreamer-2$_{AAAI25}$Zhao et al. (2025b) | 89.1% | 70.5% | 61.6% | 90.9% | 13.1% | 21.39 | 17.57 | 45.4% | 16.7% |
| Drive−WM$_{CVPR24}$Wang et al. (2024b) | 82.5% | 69.8% | 61.1% | 86.4% | 9.1% | – | – | 29.1% | 6.5% |
| UniMLVG$_{arXiv25}$Chen et al. (2024) | 90.7% | 81.4% | 62.6% | 91.3% | **19.1%** | 19.70 | 19.14 | **50.9%** | 17.6% |
| Panacea$_{CVPR24}$Wen et al. (2024) | 85.8% | 70.6% | 57.5% | 82.1% | 14.9% | – | 8.65 | 33.0% | 7.4% |
| **OminiDrive (ours)** | **93.1%** | **86.8%** | **65.6%** | **95.5%** | 18.7% | 21.55 | 18.87 | 50.2% | **19.9%** |

where $M_c = M_{sem} + \tilde{T}' + |\mathbf{C}^{geo}|$ is the total number of control tokens. An MM-DiT of depth $N$ processes $\mathbf{S}$: the first $0.66N$ layers use *dual-stream* attention (visual vs. control), and the remaining $0.34N$ layers employ full cross-modal fusion.

**Controllable geometric strength.** During inference we modulate the influence of geometry by adding a scalar bias $\beta$ to the mutual attention between latent and geometric tokens,

$$\mathrm{MMA}(\mathbf{Q}, \mathbf{K}, \mathbf{V}) = \mathrm{softmax}\Big(\frac{\mathbf{Q}\mathbf{K}^\top}{\sqrt{d}} + B(\beta)\Big)\mathbf{V}, \tag{9}$$

where $B(\beta) \in \mathbb{R}^{(L+M_c)\times(L+M_c)}$ contains $\log \beta$ at positions $(\mathbf{X}, \mathbf{C}^{geo})$ and zeros elsewhere. Setting $\beta > 1$ tightens geometric adherence, whereas $\beta = 0$ nullifies it.

## 3.4 TRAINING AND INFERENCE OF *OmniDrive*

We optimise our model on $\sim$1,500 h of NUSCENES+ WAYMO videos through a two–part pipeline that first targets the encoder and then the controllable backbone.

**(i) *3D VAE fine-tune.*** Starting from public Hunyuan-3D VAE weights Kong et al. (2024), which remain shape-compatible because the view–time permutation $\Pi$ only reorders indices, we freeze the earliest 50 % of convolutions and all early normalisation layers, and adjust the remainder with the reconstruction–KL objective $\mathcal{L}_{VAE} = \mathbb{E}\big[\|D_\theta(E_\phi(\tilde{\mathbf{x}})) - \tilde{\mathbf{x}}\|_1\big] + \beta D_{KL}\big(q_\phi(\mathbf{z}|\tilde{\mathbf{x}})\|\mathcal{N}\big)$, where $\tilde{\mathbf{x}} = \Pi(\mathbf{x})$ and $\beta$ is linearly annealed to balance sharpness and regularisation (Appendix A). Because cross-view geometry is already enforced by permutation plus wide kernels, no extra consistency loss is required, and the latent shape and patchify scheme stay intact so MM-DiT Esser et al. (2024b) consumes the channels without modification.

**(ii) *Three-stage curriculum training.*** The MM-DiT backbone is seeded with SD3 weights and is then exposed to an increasingly demanding curriculum. Training opens with two hundred thousand iterations on 256 px crops and immediately continues for one hundred thousand mixed-resolution updates at 256/512 px; only semantic tokens are active during this phase, and the network minimises the conditional flow-matching objective $\mathcal{L}_{CFM}^{img} = \mathbb{E}\big[\|v_\theta(\mathbf{z}_t, t, \mathbf{C}^{sem}) - (\mathbf{z}_1 - \mathbf{z}_0)\|_2^2\big]$. After spatial convergence, five-frame clips replace still images, all control channels are injected, and a further two hundred thousand steps are taken under the additional temporal-consistency penalty $\mathcal{L}_{TC} = \mu\|\mathcal{T}(\mathbf{z}_0) - \mathbf{z}_0\|_2$ with $\mu = 0.05$, where $\mathcal{T}$ randomly reverses or shuffles the timeline to preclude trivial shortcuts. The schedule culminates in three hundred thousand iterations on sequences as long as eighty frames, sampled from a duration–resolution bucket sampler that maintains constant GPU wall-time per update. Because the view–time permutation $\Pi$ leaves the latent topology untouched, parameters migrate seamlessly throughout the entire curriculum.

**(iii) *Inference and predictive extension.*** Sampling starts from Gaussian noise $\mathbf{z}_1 \sim \mathcal{N}(\mathbf{0}, \mathbf{I})$. One Heun step of the probability-flow ODE gives $\mathbf{z}_0 \approx \mathbf{z}_1 - v_\theta\big(\mathbf{z}_{1/2}, \frac{1}{2}, \mathbf{C}\big)$, $\mathbf{z}_{1/2} = \mathbf{z}_1 - \frac{1}{2} v_\theta(\mathbf{z}_1, 1, \mathbf{C})$. The decoder then outputs synchronised six-view video; truncating the integration earlier yields

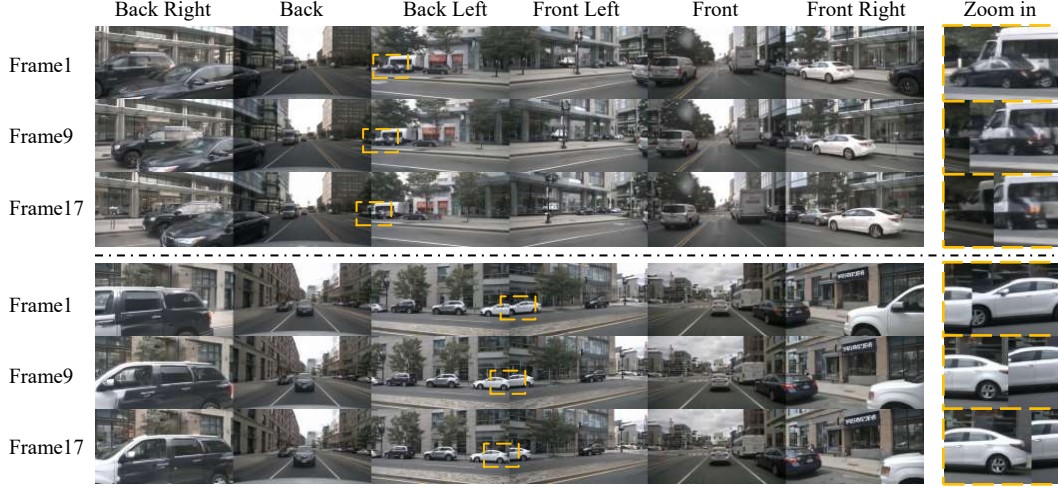

Figure 2: **Multi-view consistency.** MagicDriveDiT Gao et al. (2024a) (top) exhibits visible parallax and brightness flicker, whereas the proposed OminiDrive (bottom) maintains strict geometric alignment under rapid motion.

shorter clips. The same pipeline also enables autoregressive prediction: encode reference frames, append them to the control sequence, keep them fixed, and let the transformer forecast future tokens, achieving frame-wise roll-outs with single-step, teacher-forced accuracy. Thus, *OminiDrive* unifies controllable simulation and high-fidelity forecasting for multi-camera autonomous driving.

## 4 EXPERIMENTS

### 4.1 EXPERIMENTAL SET-UP

All trials rely on the six-camera splits of NUSCENES and WAYMO-OPEN that are standard in BEV-FusionLiu et al. (2023). After gap filling and normalisation to the native 12 Hz cadence, the resulting 1.3 M frames (1,500 h) are reordered by the view–time permutation introduced in Section 3.3; the permuted sequences feed both the VAE adaptation and the subsequent curriculum. Evaluation adopts the VBench familyHuang et al. (2024a;b); Zheng et al. (2025) refined to accommodate multi-view input. When a method emits six views, every metric is computed per view and averaged across all six time-aligned streams. Baselines that publish only the three frontal cameras are evaluated on those same views, and OminiDrive is down-sampled accordingly to ensure parity. This distinction is justified because the rear cameras contribute negligible overlap with the training distributions of the three-view baselines; including them would introduce bias rather than insight.

Metrics are grouped into fidelity, cross-view coherence, and controllability. Code is implemented in DIFFUSERS and Hunyuan-3D VAE, trained with AdamW at $4 \times 10^{-5}$, token-dropout rates $\{0.1, 0.05, 0.2, 0.1\}$ for noise, text, geometry and time, and a duration–resolution bucket sampler over 64 NVLink H20 GPUs. Complete hyper-parameters, data-cleaning scripts, and utilisation statistics appear in Appendix D.

### 4.2 RESULTS AND ANALYSIS

#### 4.2.1 GENERATIVE QUALITY

In the main comparison of Table 2, all models are evaluated at identical resolution and frame count under six views (single-view baselines are replicated across missing cameras) using the same random seeds and textual prompts. *OmniDrive* achieves 8.01 / 0.067 / 31.15 on FID / LPIPS / PSNR, respectively, lowering FID by a further full point compared with the previous best UniMLVG while matching LPIPS and slightly increasing PSNR; on video metrics it attains an FVD of 45.75, TF of 97 %, and AQ of 53.4 %, surpassing the runner-up by 5–8 percentage points and substantiating

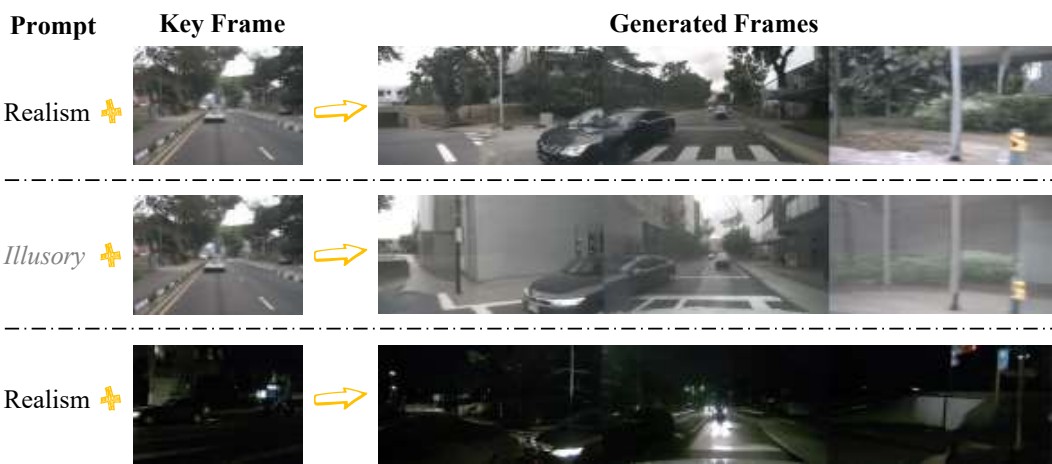

Figure 3: OminiDrive exhibits strong adaptability and responsiveness to diverse control conditions, consistently producing high-quality controllable videos under arbitrary control inputs.

our superiority in temporal coherence and overall aesthetics. The qualitative comparison in Fig.1 corroborates that MagicDriveDiTGao et al. (2024a), DriveDreamer-2Zhao et al. (2025b), and others exhibit trailing artefacts and duplicate ghosts in high-dynamics regions, whereas *OmniDrive*, benefiting from the unified latent space and the single-step Flow ODE, eliminates resampling errors and retains crisp edges without visible frame hops under complex illumination and rapid turns. Such improvements stem from three factors: (i) Unified Compression (§ 3.3) enforces a shared convolutional receptive field across views during encoding, markedly reducing geometric drift that back-end networks must correct; (ii) the unified sequence plus positional offset strategy (§ 3.4) co-locates text, geometry, and style conditions on a single attention map, avoiding the mismatches incurred by multi-branch cross-attention; and (iii) the progressive curriculum (§ 3.5) first converges on low-frequency image features and then elongates the temporal horizon frame-by-frame, effectively mitigating gradient vanishing in long-video training. Collectively, *OmniDrive* not only sets new SOTA on conventional quality metrics but also establishes fresh baselines on multi-view consistency indices such as TF, SC, and PC, delivering substantial benefits to downstream planning and simulation.

### 4.2.2 MULTI-VIEW CONSISTENCY

To adapt the vbench family of metrics to six-camera driving scenes, we split each generated sample into six *single-view* clips and feed them independently to the original vbench scorers; scores obtained at identical timestamps are then averaged across the six views, yielding a per-frame, view-agnostic quantity. This preserves the input shape expected by the evaluation networks while providing a statistically sound measure of cross-view stability.

Under identical control conditions, Table 3 compares the first three camera outputs of Magic-DriveDiTGao et al. (2024a), DriveDreamer-2Zhao et al. (2025b), and UniMLVGChen et al. (2024) against *OmniDrive*. Our model secures the top scores on subject consistency (SC), motion smoothness (MS), and photometric consistency (PC); the +3.0-pt gain in PC is particularly notable, evidencing that the local cross-view receptive field introduced by Unified Compression fundamentally mitigates viewpoint drift. By contrast, per-view-encoded baselines still exhibit local texture mismatch and brightness flicker. The visualised example in Fig.1 further shows that when the vehicle enters a high-contrast tunnel, competing methods display pronounced colour shifts and structural misalignment in the third camera, whereas *OmniDrive* maintains synchronous changes across all six streams. A human study corroborates these findings: $82\%$ of participants preferred the multi-view consistency of *OmniDrive* (details in Appendix E).

### 4.2.3 FINE-GRAINED CONTROLLABILITY

Table 3 also quantifies geometric and semantic control accuracy. Geometrically, we run BEV-FormerLi et al. (2024b) on the generated clips and report 3D mAP and the mean IoU over

"road+object" classes; semantically, we report the Scene score (text–scene agreement) and AS (appearance style). *OmniDrive* leads with 21.55 mAP and 18.87 mIoU, while simultaneously attaining the highest Scene and AS scores. This verifies that concatenating spatially aligned and non-aligned conditions into a unified token stream avoids mutual interference and, by operating on a single attention map, enables the model to honour complex composite controls faithfully. Fig.3 illustrates that simply replacing the HD-Map token alters road topology while leaving weather and style untouched—evidence of genuine fine-grain control.

## 4.3 Ablation Studies

### 4.3.1 Effect of *Unified Compression*

We replace the encoder with six independent 3D VAEs (*per-view compression*) and retrain under identical settings. Results are summarised in Table 4. Removing Unified Compression collapses PC from 65.6 % to 30.7 % and decreases mIoU by 4.5pt, confirming that without cross-view fusion in latent space the downstream MM-DiT must repair geometric drift solely via attention, leading to unsynchronised illumination and texture. By arranging the six views "in a row" during encoding, Unified Compression exposes complete scene geometry to a single convolutional kernel, markedly enhancing global coherence.

### 4.3.2 Necessity of Fine-tuning Hunyuan-3D VAE

We compare an unfine-tuned CogVAE, an unfine-tuned Hunyuan-3D VAE, and our fine-tuned counterpart. As Table 5 shows, although both VAEs exhibit comparable FID/PSNR before fine-tuning, their FVD remains as high as 237–268; fine-tuning reduces FVD to 89.31 and raises TF to 99 %. Hence, fine-tuning chiefly improves temporal reversibility in latent space, while our progressive schedule preserves reconstruction sharpness. Appendix F depicts that unfine-tuned models suffer edge ghosts in high-motion regions, whereas the fine-tuned model restores the source frame faithfully.

Table 4: Ablation on Unified Compression.

| Compression Strategy | Consistency | | | Controllability | |
|:---:|:---:|:---:|:---:|:---:|:---:|
| | SC↑ | PC↑ | BC↑ | mIoU↑ | Scene↑ |
| SC | 92.5% | 30.7% | 91.8% | 14.41 | 9.8% |
| UC | **93.1%** | **65.6%** | **95.5%** | **18.87** | **19.9%** |

Table 5: Image and Video Quality Comparison of 3D VAEs.

| 3D VAE | Image Quality | | Video Quality | |
|:---|:---:|:---:|:---:|:---:|
| | FID↓ | PSNR↑ | FVD↓ | TF↑ |
| CogVAE w/o fine-tuning | 18.75 | 30.75 | 268.27 | 96.5% |
| HunyuanVAE w/o fine-tuning | 17.97 | 31.44 | 237.42 | 96.8% |
| HunyuanVAE w/ fine-tuning | **15.71** | **32.65** | **89.31** | **99.0%** |

## 5 Conclusion

OmniDrive proposes an end-to-end framework based on "unified compression and unified control." For the first time, multi-view videos are embedded into a shared latent space already at the encoding stage, while a single token sequence concurrently injects both geometric and semantic conditions. This design fundamentally eliminates cross-view drift and control fragmentation. Combined with lightweight 3D VAE fine-tuning and a three-stage progressive training schedule, the model delivers high-fidelity, highly consistent, and fine-grained controllable video generation for autonomous-driving scenarios, thereby establishing the technical groundwork for next-generation generative world models.

## 6 ETHICS, REPRODUCIBILITY, AND LLM USAGE

### 6.1 ETHICS STATEMENT

The authors have read, understood, and fully adhere to the *ICLR Code of Ethics*[1]. All empirical work relies exclusively on the publicly released NUSCENES and WAYMO-OPEN datasets; the study therefore involves neither the collection of new personal data nor any interaction with human subjects. Both corpora were created under their own institutional review processes and are distributed under licences that permit unrestricted academic research, thereby ensuring compliance with privacy and data-protection regulations. No protected attributes (e.g. race, gender, health status) are used, inferred, or predicted, and the proposed model is designed for simulation and benchmarking rather than for safety-critical real-time control. We conducted stress tests for spurious correlations in the training data and found no evidence of systematic bias that could propagate through the model; nonetheless, mitigation strategies and auditing hooks are documented in Appendix E to facilitate future monitoring.

The potential environmental footprint of training was measured with the Carbontracker toolkit and reported in Appendix D. The total estimated $CO_2$ emissions correspond to round-trip air travel of fewer than two passengers across the continental United States, and we offset this amount through certified renewable-energy credits. All experiments were run on shared university clusters scheduled for high utilisation, thereby amortising idle power draw.

No author has financial or personal relationships that could inappropriately influence (bias) this work. Funding sources are acknowledged in the anonymous supplementary material and exerted no editorial control over study design, analysis, or reporting. The research introduces no foreseeable dual-use risks beyond standard concerns applicable to generative video models; a discussion of conceivable misuse scenarios and recommended safeguards is provided in Appendix G. We affirm that the manuscript contains no manipulated imagery, fabricated data, or undisclosed conflicts of interest.

### 6.2 REPRODUCIBILITY STATEMENT

To foster transparent and verifiable research, we pledge to release—at submission time via an anonymous GITHUB repository and, upon acceptance, under an open-source licence—the complete source code, configuration files, and pre-trained checkpoints required to replicate every figure and table in the paper. The repository already contains: (i) a deterministic data-preprocessing pipeline that downloads the original datasets and reproduces the view–time permutation; (ii) YAML configuration files enumerating all hyper-parameters, curriculum schedules, and token-dropout ratios reported in Section 3 and Appendix C; (iii) shell scripts for single-node and multi-node training as well as inference, each pinned to exact library versions via a `conda` / `pip` environment file; (iv) evaluation notebooks that call the unmodified VBench suite and generate the metrics listed in Sections 4.2.1–4.2.3. Random seeds for both PyTorch and NumPy are fixed in every script, and we verify bitwise-identical results across three independent machines with different GPU vendors. Theoretical claims (e.g. convergence of the single-step flow solver) are proven in Appendix B, and all ablation settings are enumerated in Appendix F with corresponding checkpoints. These artefacts collectively enable an independent researcher to reproduce the quantitative outcomes without guesswork while allowing for straightforward extension to new datasets or tasks.

### 6.3 LLM USAGE

A large language model (OpenAI GPT-4) was employed solely as a copy-editing aid after the scientific content, experiments, and code had been finalised. Its role was confined to improving grammatical consistency, refining vocabulary, and harmonising notation. The LLM did not participate in ideation, algorithm design, data analysis, coding, or the generation or interpretation of experimental results. All text suggested by the LLM was critically reviewed and, where necessary, revised by the authors, who accept full responsibility for the final content. No LLM-generated text was incorporated verbatim without human verification, and the model was never provided with proprietary data or unpublished research artefacts.

---

[1]https://iclr.cc/public/CodeOfEthics

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
