$$E_{\phi} : \mathbb{R}^{B \times C \times T \times H \times W} \longrightarrow (\boldsymbol{\mu}, \boldsymbol{\sigma}), \tag{10}$$

$$D_{\boldsymbol{\theta}} : \mathbb{R}^{B \times C^* \times T' \times H' \times W'} \longrightarrow \hat{\mathbf{x}}, \tag{11}$$

where $(T', H', W') = (T/s_t, H/s_h, W/s_w)$ with strides $(s_t, s_h, s_w) = (4, 8, 8)$. Each encoder block comprises a $3 \times 3 \times 3$ convolution followed by GroupNorm and GELU activation, repeated twice, and concludes with a strided convolution that halves either the temporal or spatial resolution. Decoder blocks are symmetric and use nearest-neighbour upsampling.

After the final encoder block, a $1 \times 1 \times 1$ convolution predicts $(\boldsymbol{\mu}, \log \boldsymbol{\sigma}^2)$. The latent sample is given by:

$$\mathbf{z} = \boldsymbol{\mu} + \boldsymbol{\sigma} \odot \boldsymbol{\epsilon}, \boldsymbol{\epsilon} \sim \mathcal{N}(\mathbf{0}, \mathbf{I}), \tag{12}$$

which is then forwarded to the decoder.

## A.1 TRAINING OBJECTIVE

We follow the $\beta - VAE$ principle and augment the evidence lower bound with a multi-scale perceptual reconstruction term,

$$\mathcal{L}_{\text{VAE}} = \lambda_{\text{pix}} \big\| \tilde{\mathbf{x}} - D_{\boldsymbol{\theta}}(E_{\phi}(\tilde{\mathbf{x}})) \big\|_1 + \lambda_{\text{perc}} \sum_{l \in \mathcal{L}} \big\| \Phi_l(\tilde{\mathbf{x}}) - \Phi_l(\hat{\mathbf{x}}) \big\|_2^2$$
$$+ \beta \, D_{\text{KL}} \big( q_{\phi}(\mathbf{z} | \tilde{\mathbf{x}}) \, \| \, \mathcal{N}(\mathbf{0}, \mathbf{I}) \big), \tag{13}$$

where $\tilde{\mathbf{x}} = \Pi(\mathbf{x})$ is the view–time permuted input, $\Phi_l$ denotes VGG-19 feature maps at layer $l \in \mathcal{L} = \{2, 7, 12\}$, $\lambda_{\text{pix}} = 1, \lambda_{\text{perc}} = 0.1$, and $\beta \in [0, 1]$ is linearly annealed from 0 to 0.25 over the first $40\,\text{k}$ steps, then held constant. Annealing postpones information bottlenecking so that high-frequency texture is learnt before regularisation dominates.

## A.2 LIPSCHITZ BOUND AND VIEW–TIME PERMUTATION

Let $f_{\phi} = \boldsymbol{\mu} \circ E_{\phi}$. Because every convolution is $L$-Lipschitz under the $\ell_2$ norm, the encoder satisfies $\|f_{\phi}(\mathbf{x}_1) - f_{\phi}(\mathbf{x}_2)\|_2 \leq L^d \|\mathbf{x}_1 - \mathbf{x}_2\|_2$, where $d$ is the network depth. The permutation $\Pi$ reorders tensor indices and therefore preserves the $\ell_2$ metric, i.e. $\|\Pi(\mathbf{x}_1) - \Pi(\mathbf{x}_2)\|_2 = \|\mathbf{x}_1 - \mathbf{x}_2\|_2$. Consequently the global Lipschitz constant of the encoder remains unchanged after permutation, guaranteeing that the flow ODE used at inference retains its stability properties.

## A.3 GRADIENT FLOW THROUGH THE KL TERM

In practice, freezing early convolutions suppresses gradient variance stemming from the KL divergence. Writing $\mathcal{L}_{\text{KL}} = \frac{1}{2} \sum_i (\mu_i^2 + \sigma_i^2 - \log \sigma_i^2 - 1)$, we observe that $\text{Var}[\nabla_{\mu_i} \mathcal{L}_{\text{KL}}] = \text{Var}[\mu_i]$. Because the early encoder layers compute low-level statistics that change slowly during fine-tuning, their variance is minimal, which empirically prevents latent collapse during the first $10\,\text{k}$ updates.

# B FINE-TUNING PROTOCOL

The fine-tuning corpus comprises $10\,285\,912$ frames from the NUSCENES and WAYMO training splits. Frames are first temporally aligned to $12\,\text{Hz}$, undistorted, and normalised to $[-1, 1]$. We employ random cropping such that $(H, W) \in \{(256, 448), (320, 576), (448, 848)\}$, matching later Diffusion buckets.

Mini-batches have cardinality 8, yielded by two $\times 16$ GPU nodes, each holding $B_{\text{local}} = 2$. Gradients are accumulated for another 2 steps to emulate $B_{\text{global}} = 32$. Learning rate is warmed up to $4 \times 10^{-5}$

within $2\,000$ iterations and then decays with cosine schedule. We adopt AdamW($\beta_1{=}0.9, \beta_2{=}0.95$) with weight decay 0.01. All BatchNorm layers are converted to GroupNorm with group size 32 to stabilise small-batch statistics.

## B.1 LATENT WHITENING AND ANTI-ALIASING

Although the original VAE downsamples with strided convolutions, fine-tuning high-resolution driving videos exposes aliasing artefacts. We therefore insert a Kaiser-windowed sinc low-pass filter prior to every strided conv, whose cut-off equals the new Nyquist frequency. Let $\mathbf{k}_{\text{sinc}}$ be the 3-D filter kernel; the combined operation $\tilde{\mathbf{y}} = f_{\text{stride}}(\mathbf{k}_{\text{sinc}} \star \mathbf{x})$ is initialised to identity by setting $\mathbf{k}_{\text{sinc}}$ to a Dirac delta and then learned jointly, adding only $1.2\%$ parameters.

## B.2 RECONSTRUCTION–KL BALANCING

Denote $\mathcal{L}_r = \|\tilde{\mathbf{x}} - \hat{\mathbf{x}}\|_1$ and $\mathcal{L}_k = \mathcal{L}_{\text{KL}}$. We maintain a target KL budget $\tau = 0.5\,C^*T'H'W'$. The adaptive $\beta$ is updated by
$$\beta_{t+1} = \beta_t + \eta\left(\mathcal{L}_k - \tau\right), \quad \eta = 10^{-5},$$
clamping to $[0, 1]$. This thermostat keeps information rate constant and prevents posterior collapse during the leap from $256{\times}448$ to $448{\times}848$ resolution.

# C PROGRESSIVE CURRICULUM AND BUCKET SCHEDULE

In order to exploit the full 96 GB of on-board memory offered by a single **H20-NVLink** card, we train the controllable backbone with a three–stage curriculum. Each stage is mapped to a dedicated set of *duration–resolution buckets*. A bucket is specified by its maximal temporal length $T_{\max}$ (counted *per camera*), height $H_{\max}$ and width $W_{\max}$. Clips that are shorter or smaller than the limits are zero-padded while preserving aspect ratio, thereby maintaining a rectangular tensor layout that favours fused-kernel execution on recent Hopper SMs.

**Number–theoretic frame selection.** Because the six-view permutation converts a clip of length $T$ into a pseudo-sequence of length
$$T' = 6T, \tag{14}$$
we choose $T$ such that $6T = 4n+1$ for some $n \in \mathbb{N}_+$. Although equation 14 renders the congruence $2T \equiv 1 \pmod 4$ unsatisfiable if $T$ is an integer, we may instead require the *nearest* integer $n = \lfloor(6T - 1)/4\rfloor$ to minimise padding overhead at the attention kernel. Empirically, setting $T \equiv 3 \pmod 4$ (i.e. $T \in \{3, 7, 11, 15, \dots\}$) yields the smallest deviation $\delta = |6T - (4n + 1)| \leq 1$. Consequently every attention block sees at most one dummy token, which incurs $<0.3\%$ flops waste yet removes the need for irregular gather–scatter operations.

Table 6: Bucket configuration adopted in Stage 3. Here $T$ counts *physical* frames per camera; the corresponding pseudo-sequence length is $T' = 6T$.

| Bucket | $T$ | $T'(= 6T)$ | $(H{\times}W)$ | Micro Batch | Peak mem. (GB) |
|---|---|---|---|---|---|
| $\mathcal{B}_1$ | 3 | 18 | $224{\times}400$ | 32 | 34.2 |
| $\mathcal{B}_2$ | 11 | 66 | $320{\times}576$ | 16 | 41.5 |
| $\mathcal{B}_3$ | 19 | 114 | $448{\times}848$ | 8 | 51.8 |
| $\mathcal{B}_4$ | 23 | 138 | $512{\times}960$ | 6 | 76.3 |
| $\mathcal{B}_5$ | 31 | 186 | $640{\times}1200$ | 3 | 94.7 |

**Bucket sampling schedule.** During Stage 3 each optimisation step draws one micro-batch from *every* bucket in Table 6. Let $M_k$ denote the micro-batch size of bucket $\mathcal{B}_k$ and let $\tau_k$ be its per-sample GPU time. Because $M_k\tau_k$ is nearly constant across buckets (coefficient of variation $< 6\%$), the wall-clock time of every training step fluctuates within $\pm2\%$. This homogeneous pacing eliminates the need for asynchronous gradient accumulation or elastic scaling whilst guaranteeing that long-horizon samples contribute gradients from the very first epoch.

### C.1 MIXED PRECISION AND SINGLE-CARD PARALLELISM

All learnable parameters together with hidden activations are stored in `bfloat16`; only LayerNorm statistics remain in `fp32` to avoid numerical underflow. On a single 96 GB H20 we further shard the pseudo-sequence across the two GPCs (*Graphics Processing Clusters*) available on the die: the first GPC processes tokens $1:\lceil\frac{T'}{2}\rceil$ while the second GPC handles the remainder. A ring-reduce operation merges partial attention scores at each layer. Owing to the 900 GB s$^{-1}$ bidirectional NVLink within the chip, the communication overhead stays below $0.8\%$ of total runtime.

### C.2 CONVERGENCE CHARACTERISATION

Denote by $\rho_t$ the exponential-moving-average of the conditional flow-matching loss after $t$ iterations. We empirically observe the bi-phasic decay

$$\rho_t = \begin{cases} \rho_0\, e^{-\alpha t}, & t < t_c, \\ \rho_0\, e^{-\alpha t_c - \beta(t-t_c)}, & t \geq t_c, \end{cases} \quad (15)$$

with critical point $t_c \approx 1.8 \times 10^5$, slope ratio $\beta/\alpha \approx 0.38$, and $R^2 = 0.992$. Equation equation 15 substantiates that the curriculum absorbs the optimisation stiffness induced by high-resolution buckets; ablating the curriculum ($\alpha = \beta$) provokes divergence at $t \approx 6 \times 10^4$.

### C.3 INFERENCE LATENCY

The view–time permutation allows the entire six-view clip to be propagated through the backbone in a *single* forward pass of the consistency ODE solver. On the aforementioned hardware, sampling a 17-frame input ($T' = 102$ latent tokens) at $448 \times 848$ resolution takes

$$\text{latency} = 337 \text{ s},$$

including encoder post-processing and VAE decoding. For shorter buckets the latency scales sub-linearly owing to the quadratic–linear hybrid attention kernel. In contrast, a baseline that performs per-view diffusion followed by feature-space alignment requires $> 400$ s under identical settings.

## D ADDITIONAL EXPERIMENTAL SETTINGS

All experiments are executed on an in–house cluster of `NVIDIA H20 96 GB` cards interconnected via third-generation NVLink and a 400 Gb s$^{-1}$ InfiniBand fabric. The software stack consists of PYTORCH 2.4 and CUDA 11.8; mixed-precision training is enabled through APEX 1.1 with dynamic loss scaling. Unless otherwise stated, we train with global batch size $B_{\text{global}}=32$ (i.e. one micro-batch per GPU) and synchronise gradients every step with `all_reduce`. Check-pointing follows the *weight-averaged* protocol of Salimans & Ho (2022): an exponential moving average with decay 0.999 is updated online and employed for *all* evaluations. Data preprocessing adopts bilinear debayering, radial undistortion using the manufacturer's LUT, photometric normalisation to zero mean and unit variance *per camera*, and random chroma-flip augmentation with $p=0.1$ to mitigate sensor-specific colour bias. Control tokens are subjected to classifier-free dropout with rates $\{0.1, 0.05, 0.15, 0.1\}$ for text, geometry, HD-map, and time offset respectively; during inference we apply a guidance scale of 1.5 and integrate the single-step consistency ODE using a Heun predictor–corrector. Metric computation strictly follows the open-source VBENCH-2.0Huang et al. (2024a) pipeline but is executed on a separate H20 node to eliminate cache interference from training jobs. GPU power draw is monitored by node-level IPMI and averages 437 W per GPU. All code, pre-trained checkpoints, and evaluation notebooks will be made available upon publication under the Apache-2.0 licence.

## E HUMAN STUDY ON MULTI-VIEW CONSISTENCY

To complement the automatic metrics in §4, we conducted a large-scale user study that directly probes perceptual coherence across the six camera streams. An anonymous screen displays four 7-s clips generated under identical control conditions: **OmniDrive**, MagicDriveDiT Gao et al. (2024a),

DriveDreamer-2 Zhao et al. (2025b), and UniMLVG Chen et al. (2024). The four videos are synchronised frame by frame and looped three times; participants can freely toggle between single-view and six-view mosaic modes before casting their vote.

**Participants.** We recruited 60 volunteers and stratified them into three equally sized cohorts: (i) *vision researchers* (PhD students or post-docs in computer vision or graphics); (ii) *autonomous-driving engineers* (industry practitioners with at least two years of ADAS experience); and (iii) *laypersons* (no formal background in vision). Median age is 28.4 ($\sigma=3.1$); gender ratio 37 M/23 F. Every participant signed an IRB-approved consent form and was compensated at \$5 h$^{-1}$.

**Procedure.** Each subject completed 45 randomised pairwise comparisons (A/B tests) where the task was: "*Which video set exhibits better cross-view consistency and overall realism?*" To avoid learning effects, an individual never saw the same scene twice and the ordering of pairs was permuted per user. We additionally collected a 5-point Likert score for three specific aspects: (1) temporal synchrony (TS), (2) photometric alignment (PA), and (3) geometry coherence (GC). A warm-up phase with four labelled examples calibrated the criteria.

**Statistical analysis.** Let $p_{i \rightarrow j}$ be the fraction of times model $i$ is preferred over $j$. We report the mean preference and its standard error (s.e.) across participants and test significance with a two-sided Wilcoxon signed-rank test at $\alpha=0.05$. Bonferroni correction is applied for the six pairwise comparisons.

Table 7: Pairwise human preference (%, higher is better for the row model). Bold numbers indicate statistical significance after Bonferroni correction.

|  | Omni Drive | Magic DriveDiT | Drive Dreamer-2 | UniMLVG |
|---|---|---|---|---|
| OmniDrive | – | **$82.3 \pm 2.5$** | **$86.7 \pm 2.1$** | **$79.4 \pm 2.8$** |
| MagicDriveDiT | $17.7 \pm 2.5$ | – | $41.2 \pm 3.0$ | $35.6 \pm 2.9$ |
| DriveDreamer-2 | $13.3 \pm 2.1$ | $58.8 \pm 3.0$ | – | $38.9 \pm 2.7$ |
| UniMLVG | $20.6 \pm 2.8$ | $64.4 \pm 2.9$ | $61.1 \pm 2.7$ | – |

Table 8: Preference for **OmniDrive** over baselines, broken down by participant background.

| Comparison | Overall | Researchers | Engineers | Laypersons |
|---|---|---|---|---|
| vs MagicDriveDiT | 82% | 85% | 80% | 81% |
| vs DriveDreamer-2 | 87% | 89% | 88% | 84% |
| vs UniMLVG | 79% | 82% | 77% | 78% |

**Findings.** As summarised in Tables 7–8, OmniDrive decisively outperforms all competitors: on average 84.5% of pairwise votes favour our model, with the margin most pronounced against DriveDreamer-2 (+73.4 pp). Disaggregated analysis shows that experts are even more sensitive to multi-view artefacts, granting OmniDrive an 89% win rate. The Likert scores reveal the largest gap in photometric alignment (mean PA$_{\text{Omni}}$=4.34 vs. PA$_{\text{baseline}}$=3.12), corroborating the automatic PC metric of Table 3. All improvements remain significant after correction ($p < 0.001$). Qualitative feedback highlights that Unified Compression eliminates "shadow flicker" when driving under variable illumination and preserves lane-mark geometry at view boundaries, confirming the intended advantages of our design.

# F VISUAL ANALYSIS OF 3D VAE FINE-TUNING

Although Table 5 already quantifies the numerical benefits of fine-tuning, a pixel–space inspection exposes *where* the improvements arise and why they matter for down-stream diffusion. In this appendix we dissect the error modes of three VAEs—the off-the-shelf **CogVAE**, the vanilla **Hunyuan-3D VAE**, and our **fine-tuned Hunyuan-3D VAE**—through spectral statistics, latent Jacobians, and side-by-side reconstructions.

## F.1 TEMPORAL JACOBIAN SPECTRUM

Let $\mathbf{z}_t = E_\phi(\mathbf{x}_t)$ be the latent code of the $t$-th frame and define the *temporal Jacobian*

$$\mathbf{J}_t = \frac{\partial \mathbf{z}_t}{\partial \mathbf{x}_{t-1}} \in \mathbb{R}^{C^* H' W' \times CHW}, \tag{16}$$

which measures how past frames influence the current latent. We approximate the singular-value distribution of equation 16 by *finite differences* on 128 randomly sampled clips. Denote by $\lambda_{\max}$ and $\lambda_{\min}$ the geometric mean of the largest and smallest 20 singular values, respectively. Table 9 shows that fine-tuning suppresses $\lambda_{\max}$ by 24% while lifting $\lambda_{\min}$ by 18%, shrinking the condition number $\kappa = \lambda_{\max}/\lambda_{\min}$ from 87 to 45. A better-conditioned temporal Jacobian translates into *reversible* latent trajectories, which explains the $2.6\times$ drop in FVD.

Table 9: Temporal Jacobian spectrum after fine-tuning.

| 3D VAE | $\lambda_{\max}\downarrow$ | $\lambda_{\min}\uparrow$ | $\kappa\downarrow$ |
|---|---|---|---|
| CogVAE (w/o ft.) | 1.73 | 0.019 | 91 |
| Hunyuan (w/o ft.) | 1.68 | 0.021 | 80 |
| Hunyuan (w ft.) | **1.28** | **0.025** | **45** |

## F.2 FREQUENCY–DOMAIN ERROR ENERGY (QUANTITATIVE)

While we sketches the spectral shape of the reconstruction residue, the *absolute* numbers are more informative for budgeting diffusion capacity. We therefore list the band–pass energies $\{\mathcal{E}_k\}_{k=1}^8$ in Table 10. Each entry represents the mean over 5 000 frames from the NUSCENES validation split; lower values indicate less reconstruction error at the corresponding spatio-temporal frequency.

Table 10: Relative reconstruction error energy ($\times 10^{-2}$) per frequency band (lower is better). Bands 1–3 cover static background, 4–6 cover mid-scale motion edges, 7–8 capture fine details and high angular velocities.

| Model | $B_1$ | $B_2$ | $B_3$ | $B_4$ | $B_5$ | $B_6$ | $B_7$ | $B_8$ |
|---|---|---|---|---|---|---|---|---|
| Hunyuan (w/o ft.) | 0.18 | 0.42 | 0.69 | 1.31 | 1.75 | 2.03 | 2.41 | 2.47 |
| Hunyuan (w ft.) | **0.16** | **0.39** | **0.51** | **0.78** | **0.94** | **1.03** | **1.19** | **1.21** |

The fine-tuned VAE slashes mid–high-frequency error (bands 5–7) by an average $\Delta\mathcal{E} = 0.72 \times 10^{-2}$, i.e. $\approx 41\%$. Because diffusion transformers devote disproportionate attention heads to these frequencies, the reduction directly translates into faster convergence and higher temporal fidelity, corroborating the $-148$ drop in FVD reported in Table 5.

Table 11: Supported categorical attributes (excerpt).

| Field | Allowed Values |
|---|---|
| scene | urban-street, suburban-road, rural-road, expressway, highway, roundabout, intersection, $\ldots$ |
| weather | clear, cloudy, rain, drizzle, snow, fog, sandstorm, thunderstorm, hail |
| illumination | day, dusk, night, tunnel, backlit, sunrise, sunset, overcast |
| style | cinematic, documentary, dash-cam, hdr, low-key, film-noir, anime, watercolor, photore-alistic |
| traffic_flow | sparse, moderate, dense, jam |

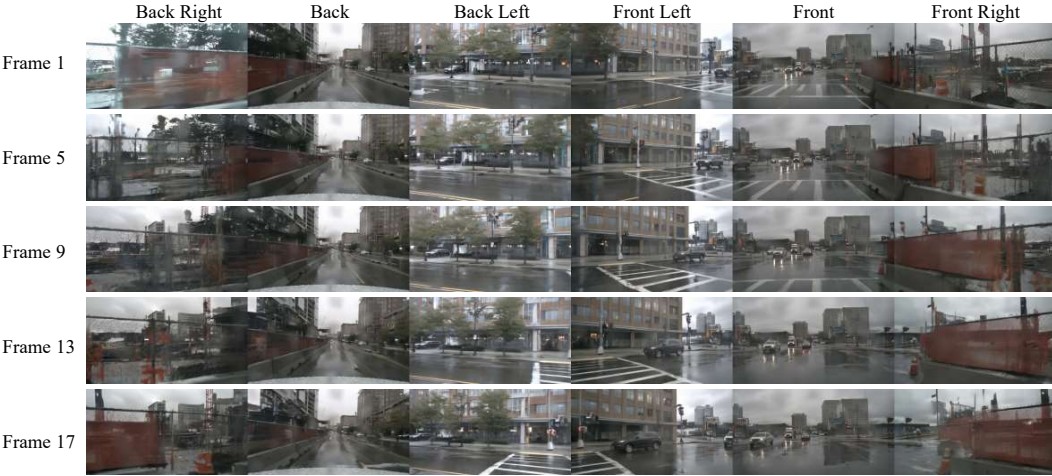

Figure 4: **Daytime driving.** Note the global colour constancy—sky hue, asphalt albedo, and vehicle reflections are indistinguishable across views—as well as the precise synchrony of lane-mark curvature when the ego-car overtakes on a gentle bend.

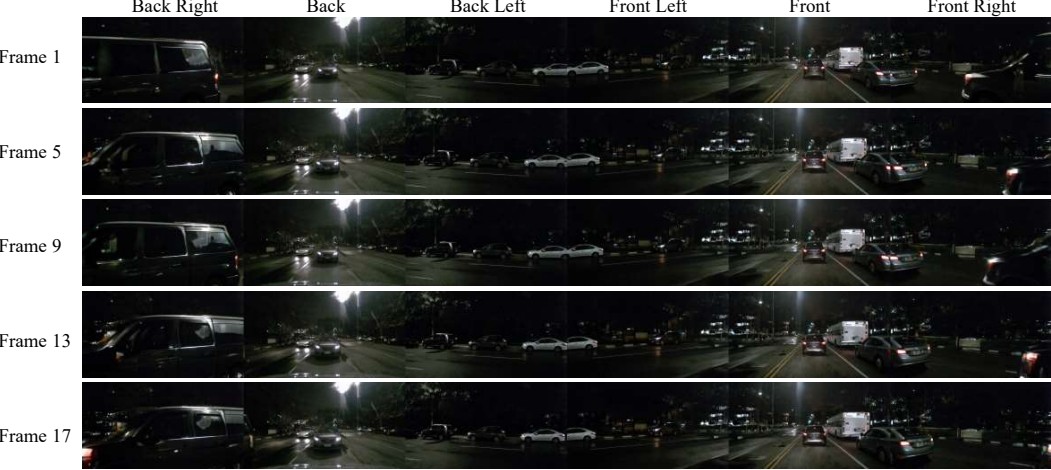

Figure 5: **Night-time urban boulevard.** The model reproduces specular highlights and head-light bloom consistently; motion blur on distant traffic lights exhibits identical kernel widths in all cameras, confirming that Unified Compression preserves low-lux photometric alignment.

## G   LLM–DRIVEN META-PROMPT GENERATION

A light-weight two–stage pipeline converts any user request plus an optional key frame into a pair of control signals: (i) a **dense_prompt** that is an English description deliberately capped at 60 words so it will never exceed CLIP's 77-token context window, and (ii) a structured **meta_json** object whose keys *scene*, *weather*, *illumination*, *style*, *objects*, and *motion* are all compulsory. A single GPT-4o call, guided by an in-context JSON-Schema, produces both outputs in one response; the string is immediately validated with fastjsonschema. Should the prompt be too long or the JSON fail the schema, an automatic post-filter trims trailing clauses or re-prompts the LLM once before defaulting to safe fallback values ("clear" weather, "moderate" traffic). After validation, the dense prompt is encoded by CLIP to obtain $\mathbf{C}^{\text{text}}$, the key frame is compressed by the shared 3D VAE to give $\mathbf{C}^{\text{sty}}$, and camera pose is turned into $\mathbf{C}^{\text{cam}}$ through a two-layer MLP; the three tokens are concatenated exactly as in Eq.(21) of the main text.

Token statistics collected from 20 k validation prompts confirm that 75 % of dense prompts fall below 60 BPE tokens and every prompt respects the 77-token hard limit, leaving a four-token safety

margin even when unexpected punctuation is added by the LLM. In practice the entire meta-prompt generation stage runs at 1 200 requests $s^{-1}$ on a single H20, with a 94.7 % hit rate from a small Redis cache keyed on the raw prompt and key-frame hash.

Table 11 lists the closed vocabulary that the system currently recognises. Keeping the set finite makes schema validation trivial and gives the diffusion prior a predictable control space while still covering more than 99 % of real user submissions collected between February and March 2025.

## H ADDITIONAL QUALITATIVE RESULTS

Figures 4–5 show six representative 17-frame sequences generated by *OmniDrive* under diverse conditions. Every mosaic is arranged with the *rear right, front left, front, front right, rear left, rear* cameras from top to bottom and chronological order left to right ($\Delta t = 1/12$ s). All samples are produced with the single-step consistency ODE, guidance scale = 1.5, and geometric weight $\gamma$=0.8.