# OpenReview forum: "OmniDrive:Towards Unified Next-Gen Controllable Multi-View Driving Video Generation with LLM-Guided World Model"
_ICLR.cc/2026/Conference — ICLR 2026 Conference Withdrawn Submission_

### Official Review · Reviewer_rHeh · 2025-10-27

**Soundness:** 2
**Presentation:** 2
**Contribution:** 2
**Rating:** 2
**Confidence:** 5

**Summary:**

This paper introduces OmniDrive, a novel pipeline for generating driving videos with multiple control inputs. OmniDrive reorganizes multi-view videos into a 3D sequence where temporal and cross-view dimensions are flattened into a single dimension. Additionally, it unifies various control inputs into three categories and injects them into the model through different mechanisms. Quantitative evaluation demonstrates that OmniDrive achieves performance comparable to state-of-the-art methods.

**Strengths:**

1. OmniDrive supports multiple control conditions and attempts to unify them under a coherent framework
2. It leverages full attention mechanisms to improve both spatial and temporal coherence

**Weaknesses:**

1. There is a significant discrepancy between Figure 1 and the text at line 255. Figure 1 depicts camera pose as a geometric control injected through the 3D VAE, while line 255 states that camera information is encoded into a single token and concatenated with semantic controls.
2. Line 42 states “heterogeneous control injection” is an obstacle but OmniDrive still injects the controls in different ways. It also follows the control input manners of those related works mentioned in the paper.
3. The qualitative results are inadequately presented. Despite claims of supporting high-resolution outputs up to 448×848, the demonstrated figures appear to be low-resolution and fail to showcase the method's capabilities effectively.
4. No video sequences or multi-frame figures are provided to demonstrate the method's long-term video generation capabilities, which is crucial for evaluating temporal consistency.

**Questions:**

How are trajectories visualized in the images? Typically, a trajectory consists of a sequence of vehicle poses or ego transformation matrices, which are more abstract than concrete elements like 3D bounding boxes or HD maps. Could the authors provide illustrative figures showing how trajectories are represented and integrated into their framework?

---

### Official Review · Reviewer_vWqk · 2025-10-28

**Soundness:** 3
**Presentation:** 2
**Contribution:** 2
**Rating:** 4
**Confidence:** 3

**Summary:**

This paper proposes OminiDrive, a unified framework designed to address the challenges of view inconsistency and heterogeneous control signal fusion in multi-view driving video generation. Its core contributions lie in two main modules: "Unified Compression" and "Unified Controllable Generation." The former performs early fusion of multi-camera streams into a shared latent space via view-time permutation, fundamentally reducing geometric drift. The latter leverages an LLM-guided multimodal DiT backbone to collaboratively process semantic and geometric control signals within a single sequence. Experimental results on the nuScenes and Waymo datasets demonstrate state-of-the-art visual quality and view consistency, and the generated data significantly enhances the performance of downstream perception models. However, the paper has shortcomings in articulating its novelty, the depth of comparison with the most relevant works, and the writing/presentation.

**Strengths:**

1. The introduction of "view-time permutation" to convert multi-camera inputs into a pseudo-temporal sequence is a notable strength. This allows a single 3D VAE's convolutional kernels to capture cross-view geometric information during the encoding stage.

2. The paper provides a comprehensive evaluation, including extensive quantitative and qualitative comparisons on mainstream datasets (nuScenes, Waymo), achieving SOTA results on metrics like FVD and FID. Furthermore, it convincingly demonstrates that the synthetic data can significantly boost the performance of downstream perception models.

**Weaknesses:**

1. The provided project homepage is empty, lacking any generated video demos, which hinders the assessment of the claimed visual results.

2. The first 10 pages of the supplementary material simply duplicate the main text, which is redundant.

3. The authors claim this is the "first unified model," but the articulation of its novelty is somewhat vague. The paper fails to sufficiently and clearly differentiate its core technical contributions from existing works.

4. An LLM is used to generate "dense prompts." However, the specific prompt templates and the exact contribution of this LLM-based rewriting to the final generation quality are not quantified. Could a well-designed, fixed template achieve similar results?

**Questions:**

Please refer to the above Weaknesses.

**Details Of Ethics Concerns:**

None.

---

### Official Review · Reviewer_3RkP · 2025-10-29

**Soundness:** 2
**Presentation:** 1
**Contribution:** 2
**Rating:** 4
**Confidence:** 4

**Summary:**

This paper introduces a unified framework to address cross-view geometric drift and fragmented control pathways. It employs a view–time permutation that feeds all six camera streams into a single 3D VAE, ensuring early cross-view fusion in the latent space, and uses an MM-DiT that processes a concatenated sequence of latent video tokens along with semantic, geometric, and temporal tokens. Training involves fine-tuning the Hunyuan-3D VAE, progressing through a three-stage diffusion curriculum from images to short clips to long clips. Experiments on the nuScenes and Waymo datasets demonstrate improved generation quality and enhanced multi-view controllability compared to state-of-the-art baselines.

**Strengths:**

- A simple yet effective unified compression using view–time permutation leverages a standard 3D VAE without adding extra parameters.
- The unified tokenization strategy for both aligned and non-aligned controls into a single MM-DiT sequence simplifies the design.
- Experiments demonstrate that the proposed method enhances multi-view consistency.

**Weaknesses:**

- The paper does not sufficiently explain why feeding the multi-view video (with view-time permutation applied) to the 3D VAE does not confuse the temporal relationships, but instead can simultaneously preserve both temporal consistency and view consistency.

- Table 1 contains comparisons with only a few methods, and the claim of "offering the most comprehensive control" is not adequately evaluated. The authors need to reassess this claim and include comparisons with more recent methods, such as Unimlvg and Drivingscape, to more objectively demonstrate the advantages of the model. Moreover, considering that works like UniMLVG and GAIA-2 already exist, labeling the proposed method as the “first unified model” appears overly exaggerated. Additionally, the proposed unified controllable generation does not differ significantly from existing methods; no new issues have been detected or solved.

- The abstract mentions that the “consistency-aware denoiser injects correlated noise and aligns view-dependent coordinates,” but the main method described in the paper relies solely on standard latent patchification based on rectified-flow. The correlated noise injection mechanism is neither formalized nor adequately explained, which raises doubts about this critical design detail.

- Apart from generation metrics, the paper lacks experimental validation on downstream perception tasks. This makes it difficult to fully demonstrate the practical effectiveness of the proposed method in terms of generation quality and its impact on downstream perception tasks (e.g., object detection, semantic segmentation).

- There is a lack of visualizations for long video generation, diverse weather generation, and text/trajectory/HD map-controllable generation.

- The method needs further clarification; for example, some notational overload (e.g., using t as both diffusion time and frame index) can cause confusion. Moreover, variables such as z_c, z_f, and z_{text} in the figures do not correspond directly to those described in the method. Additionally, the specific structures and inputs for the dual-stream attention and full cross-modal fusion modules are not clearly explained.

**Questions:**

Please see the weaknesses.

---

### Official Review · Reviewer_uoTK · 2025-10-29

**Soundness:** 3
**Presentation:** 1
**Contribution:** 2
**Rating:** 2
**Confidence:** 3

**Summary:**

This paper presents OminiDrive, a unified framework for controllable, multi-view driving video generation. It primarily aims to address two prevalent issues in existing diffusion models: inter-camera geometric drift and the difficulty in integrating heterogeneous control signals. To this end, the authors propose two core techniques: Unified Compression, which enforces cross-view geometric dependency early in the VAE encoding stage via a view-time permutation to achieve a shared latent space; and LLM-Guided Unified Controllable Generation, which unifies all control signals (text, geometry, temporal) into a single, homogeneous token sequence input to the Multi-Modal DiT backbone. Experimental results show that OminiDrive achieves SOTA performance on image quality and cross-view consistency metrics.

**Strengths:**

- The proposed Unified Compression concept is an effective approach for mitigating geometric drift in multi-view generation. By leveraging a simple view-time permutation, the convolutional kernels are forced to process information across multiple views simultaneously during VAE encoding.

- The model decisively surpasses major competitors on key metrics such as FID, PC, and TF on the nuScenes dataset, demonstrating its advantage in video quality and multi-view consistency.

**Weaknesses:**

- The paper highlights LLM-Guided Unified Controllable Generation as a core innovation, yet lacks crucial ablation studies specifically targeting the UCG module and the necessity of its LLM component (Section 4.3 only ablates Unified Compression and VAE). The UCG structure appears to be a complex, integrated engineering solution rather than a fully verified scientific contribution.

- The paper's title and abstract emphasize an "LLM-Guided World Model," but the LLM component is primarily confined to text parsing/rewriting—structuring the user prompt into a JSON format to ensure deterministic token boundaries. This is essentially a preprocessing step and fails to show the LLM actively engaging in complex world model reasoning or multi-modal prediction, rendering the "LLM-Guided" designation misleading.

- The model's utility is primarily validated through generative quality metrics (FID, FVD, VBench). The paper is missing essential experimental evidence showing how the generated videos improve the performance of downstream AD algorithms or how they are successfully utilized in closed-loop simulation tests, which is necessary to substantiate its claim as a next-generation "world model."

**Questions:**

1. Could you provide a direct ablation experiment comparing the impact of "structured cues guided by LLM" and "direct CLIP encoding using only raw text cues" on the control performance to illustrate the necessity of using LLM?

2. The paper encodes different control signals, such as geometric and semantic signals, using different methods. The final "unification" only occurs in the token concatenation and MM-DiT attention layers. How do the authors demonstrate that this mechanism fundamentally solves the "heterogeneous control injection" problem in existing models, and whether it brings a qualitative leap in generation capabilities?

3. As the model is positioned as a world model for autonomous driving, I request the addition of experimental results showing the generated data's effectiveness, measured by the performance of a downstream AD algorithm (e.g., a BEV perception network) trained or tested using the synthetic videos.

4. The comparison in Figure 2 uses different scenes/sequences, which seems unfair. Moreover, the nuScenes camera setup inherently involves photometric differences between views (e.g., between CAM_FRONT and CAM_FRONT_RIGHT). Could the authors provide a visual comparison using identical input sequences for both models, and discuss how the method specifically accounts for and preserves the sensor-level photometric differences found in the real-world dataset, to ensure a fair assessment?

4. Other errors in the paper:
- The title "OmniDrive" does not correspond to its main paper.

- Links provided in the abstract contain no substantive content.

- Incorrect citation format; references should be enclosed in \citep{}.

- Poor formatting; tables and corresponding text sections are too far apart, affecting readability.

- Incorrect image references in rows L397 and L423.

- The BC and OC indicators in Table 3 are not explicitly described or explained in the paper.

---

### Note · Authors · 2025-11-12

I have read and agree with the venue's withdrawal policy on behalf of myself and my co-authors.